# Machine Learning-Based Live Weight Estimation for Hanwoo Cow

**Changgwon Dang** [1,†] , **Taejeong Choi** [1,†] , **Seungsoo Lee** [1] , **Soohyun Lee** [1] , **Mahboob Alam** [1] , **Mina Park** [1] , **Seungkyu Han** [2] , **Jaegu Lee** [1,*] and **Duytang Hoang** [2,*]

1 National Institute of Animal Science, RDA, Cheonan 31000, Chungcheongnam-do, Korea
2 ZOOTOS Co., Ltd., R&D Center, Anyang 14118, Gyeonggi-do, Korea
* Correspondence: jindog2929@korea.kr (J.L.); buffalo@zootos.com (D.H.);
  Tel.: +82-10-4030-1929 (J.L.); +82-10-4682-0230 (D.H.)
† These authors contributed equally to this work.

**Abstract:** Live weight monitoring is an important step in Hanwoo (Korean cow) livestock farming. Direct and indirect methods are two available approaches for measuring live weight of cows in husbandry. Recently, thanks to the advances of sensor technology, data processing, and Machine Learning algorithms, the indirect weight measurement has been become more popular. This study was conducted to explore and evaluate the feasibility of machine learning algorithms in estimating the body live weight of Hanwoo cow using ten body measurements as input features. Various supervised Machine Learning algorithms, including Multilayer Perceptron, k-Nearest Neighbor, Light Gradient Boosting Machine, TabNet, and FT-Transformer, are employed to develop the models that estimate the body live weight using body measurement data. Data analysis is exploited to explore the correlation between the body size measurements (the features) and the weights (target values that need to be estimated) of cows. Data analysis results show that ten body measurements have a high correlation with the body live weight. High performance of all applied Machine Learning models was obtained. It can be concluded that estimating the body live weight of Hanwoo cow is feasible by utilizing Machine Learning algorithms. Among all of the tested algorithms, LightGBM regression demonstrates not only the best model in terms of performance, model complexity and development time.

**Keywords:** Hanwoo cow; live weight estimation; machine learning; deep learning

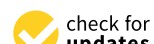



## 1. Introduction

In South Korea, among all types of beef in the market, people prefer native beef despite their prices being much higher than that of imported products. Among four types of native cattle breeds being raised for beef demand, Hanwoo is the most popular one [1]. With highly marbled fat, thin muscle fibers, and minimal content of connective tissues, Hanwoo beef is well-known for its distinctive flavor [2]. To maintain the valuable characteristics of beef, the livestock management procedure takes an important role. In that procedure, livestock live weight monitoring is critical since it is considered one of the most important traits affecting animal condition [3]. In the management procedure, accurately estimating or measuring live weight is of fundamental importance to any livestock research and development.

Currently, there are two main approaches available to measure the live weight of livestock, including direct and indirect methods. The direct measurement method using scales can get very high accuracy. However, it still has some limitations. First of all, the measurement process in this approach requires removing bulls from cages or paddocks, and guiding them one by one to the weighting station or the site of scale. This process is highly time-consuming and cumbersome. Sometimes workers who stay close to the bulls might get hurt as some cattle individuals are very stubborn. Secondly, this

process is believed to be able to cause stress and potentially harmful to bulls, even leading to weight loss or death [4]. Because of those disadvantages of the direct measurement methods, various ways of indirect measurement have been proposed as the alternative approach [4–12]. The indirect approach is considered an estimation of the true value of live weight since it indirectly computes that value using sensor data and computational techniques. Weight estimation using body size measurements has been used extensively in the livestock industry, for both carcass weight and live weight. A typical indirect live weight measurement method consists of three steps. In the first step, different body characteristics and size of cattle are collected by sensors such as 2D camera [4], thermal camera [13], 3D camera [14,15] and ultrasonic sensor [16,17]. In the second step, body features are extracted by data processing techniques. Finally, body features are fed into a regression model to estimate the body weight.

The task of estimating body weight using body measurements can be considered a regression problem where the body measurements are input features and body weight is the target value that the regression model needs to predict. The estimation of dairy Holstein cattle live weight was reported by Tedde et al. with Root Mean Square Error (RMSE) ranging from 52 to 56 kg [12]. A study on estimating the live weight of pigs was conducted by Sungirai et al. [11,18]. Regarding sheep live weight estimation, a study was conducted by Sabbioni et al. [10]. It should be noted that one estimation model, when applied to different cattle breeds could have different prediction performances. For example, regression analysis was exploited to predict body weight from body measurements in Holstein, Brown Swiss, and crossbred cattle with R2 scores of 92%, 95%, and 68%, respectively [8]. In the case of the Hanwoo cattle, the study of live weight estimation was carried out by Jang et al. with the performance demonstrated by RMSE and MAPE errors of 51.4% and 17.1%, respectively, using body size measurements including body length, withers height, chest width, and body width [7].

Machine Learning (ML) has a long history dating back to the year 1959, and the term was coined by Arthur Samuel [19]. ML algorithms can be categorized into three types of learning: reinforcement learning, unsupervised learning, and supervised learning. Among three categories of learning algorithms, supervised learning is employed for the task of estimating live body weight of cattle in livestock. Supervised learning algorithms try to learn from the labeled datasets to approximate the mapping function between inputs (features) and outputs (target values). There are a huge number of supervised learning algorithms, such as Linear Regression, k-Nearest Neighbor (kNN), Support Vector Machine (SVM), Decision Tree, and Artificial Neural Network (ANN) or Neural Network (NN). Among existing supervised learning algorithms, ANN is considered an arbitrary accuracy function approximation [20]. ANN is a ML algorithm that utilizes data computational structure inspired by the nervous system of the superior organisms [21]. A typical NN consisted of an input layer, hidden layers, and an output layer. Deep Learning (DL) or Deep Neural Network (DNN) are special neural networks that consist of many layers of data processing units. The main advantage of DNN over NN is the ability to automatically learn features from raw data, without the hand-craft features. Nowadays, DL and DNNs are dominant in almost every kind of unstructured data: serial data, 2D data, and 3D data. Various types of DNN have been proposed for different types of data. For example, Convolutional Neural Network (CNN) models are suitable for image data [22] while attention-based neural network models are dominant in natural language processing applications [23]. In tabular data, tree-based ensemble learning is still believed to outperform other types of learning algorithms [24,25]. However, some DNN models proposed recently can have comparable performance in tabular data tasks [24].

In this paper, a dataset including 33,536 samples of Hanwoo cows is used to develop live weight predictive models. Each sample consists of 10 body measurements along with the age and body weight. Although previously, live weight body estimation of Hanwoo was studied by Jang et al. [7] with only 4 body measurements, the predictive performance is still low (RMSE 51.4 and MAPE 17.1%). In this paper, a dataset with more features

and much more samples, besides conventional ML algorithms, more advanced ML-based sophisticated predictive models are employed to develop live weight predictive models. The main contributions of this paper are as follows:

- Analyze ten body measurements of Hanwoo and their impact on the prediction of body weight.
- Investigate ML algorithms in estimating live body weight.
- Improve predictive performance over previous studies.

## 2. Materials and Methods

### 2.1. Hanwoo Body Measurement Data

The Hanwoo data used in this research was provided by the National Institute of Livestock Science, Korean Rural Development Administration. The data consist of 33,546 records of male individuals with ages of 6, 12, 18, and 24 months. The total numbers of individuals in four age groups are 4088, 16,574, 7185, and 5699 respectively. The dataset is split into training, validation, and test datasets with a ratio of 70%–15%–15%. The training and validation datasets are used for developing predictive models. During the training process, the validation dataset helps avoid the over-fitting phenomenon of the training process. The test dataset is used to evaluate the performance of trained models on unseen data.

### 2.2. Body Size Measurements

Each data sample is an observation of a cow individual, consisting of ten size measurements measured in centimeters (cm), age in month, and weight in kilogram (kg). The body size measurement annotations are taken from [26]. Details are demonstrated in Figure 1 and Table 1. The statistical summary of the data is in Tables 2–5.

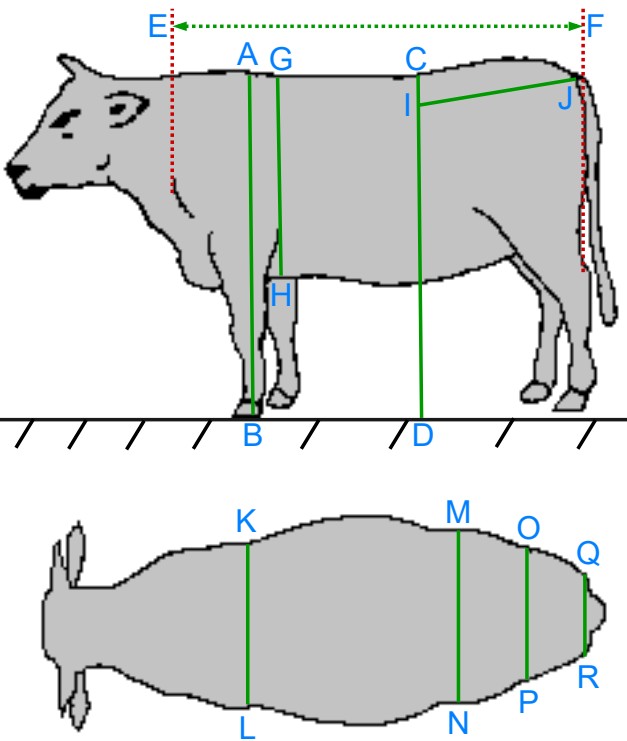

**Figure 1.** Hanwoo cow body measurements.

**Table 1.** Hanwoo cow body measurements.

| Feature Name | Description | Annotation |
|---|---|---|
| Chego | Wither height | A–B |
| Sibza | Hip height | C–D |
| Chejang | Body length | E–F |
| Hungsim | Chest depth | G–H |
| Gojang | Rump length | I–J |
| Hongpok | Chest width | K–L |
| Yogak | Rump width | M–N |
| Gonpok | Pelvic width | O–P |
| Jagol | Hip bone width | Q–R |
| Hungque | Chest girth | Circumference of G–H |

**Table 2.** Statistical summary of 6-month age cows.

| | Chego (cm) | Sibza (cm) | Chejang (cm) | Hungsim (cm) | Hungpok (cm) | Yogak (cm) | Gojang (cm) | Gonpok (cm) | Jogol (cm) | Hungwe (cm) | Weight (kg) |
|---|---|---|---|---|---|---|---|---|---|---|---|
| Mean | 102.168 | 105.003 | 108.249 | 50.309 | 29.036 | 28.914 | 36.322 | 32.225 | 19.222 | 131.345 | 183.366 |
| STD | 5.422 | 5.205 | 7.393 | 4.070 | 4.473 | 4.219 | 4.401 | 4.128 | 3.583 | 7.400 | 25.367 |
| Min | 85 | 87 | 85 | 32 | 16 | 17 | 24 | 20 | 11 | 106 | 104 |
| Q1 | 99 | 102 | 103 | 48 | 26 | 26 | 34 | 30 | 17 | 127 | 167 |
| Q2 | 102 | 105 | 108 | 50 | 29 | 28 | 36 | 32 | 19 | 132 | 184 |
| Q3 | 106 | 109 | 113 | 52 | 31 | 31 | 39 | 34 | 20 | 136 | 201 |
| Max | 119 | 123 | 132 | 65 | 47 | 42 | 50 | 45 | 32 | 155 | 266 |

**Table 3.** Statistical summary of 12-month age cows.

| | Chego (cm) | Sibza (cm) | Chejang (cm) | Hungsim (cm) | Hungpok (cm) | Yogak (cm) | Gojang (cm) | Gonpok (cm) | Jogol (cm) | Hungwe (cm) | Weight (kg) |
|---|---|---|---|---|---|---|---|---|---|---|---|
| Mean | 120.866 | 123.228 | 135.840 | 62.432 | 38.411 | 37.976 | 45.709 | 40.628 | 22.206 | 169.321 | 362.839 |
| STD | 4.424 | 4.527 | 6.708 | 2.782 | 3.762 | 2.945 | 3.217 | 3.102 | 2.641 | 8.201 | 44.940 |
| Min | 105 | 107 | 113 | 42 | 26 | 25 | 32 | 27 | 14 | 141 | 218 |
| Q1 | 118 | 120 | 132 | 61 | 36 | 36 | 44 | 39 | 20 | 164 | 332 |
| Q2 | 121 | 123 | 136 | 62 | 38 | 38 | 46 | 41 | 22 | 169 | 360 |
| Q3 | 124 | 126 | 140 | 64 | 41 | 40 | 48 | 43 | 24 | 175 | 392 |
| Max | 135 | 139 | 157 | 77 | 53 | 51 | 60 | 54 | 30 | 197 | 508 |

**Table 4.** Statistical summary of 18-month age cows.

| | Chego (cm) | Sibza (cm) | Chejang (cm) | Hungsim (cm) | Hungpok (cm) | Yogak (cm) | Gojang (cm) | Gonpok (cm) | Jogol (cm) | Hungwe (cm) | Weight (kg) |
|---|---|---|---|---|---|---|---|---|---|---|---|
| Mean | 129.598 | 130.950 | 148.816 | 70.228 | 44.386 | 44.712 | 49.506 | 45.579 | 24.536 | 193.284 | 498.297 |
| STD | 6.090 | 6.178 | 7.951 | 3.522 | 4.388 | 3.593 | 3.871 | 3.930 | 3.348 | 10.501 | 66.607 |
| Min | 110 | 111 | 124 | 59 | 31 | 32 | 38 | 34 | 16 | 159 | 296 |
| Q1 | 126 | 128 | 144 | 68 | 42 | 42 | 47 | 43 | 22 | 186 | 452 |
| Q2 | 130 | 132 | 149 | 70 | 44 | 45 | 50 | 46 | 24 | 194 | 496 |
| Q3 | 134 | 135 | 155 | 73 | 47 | 47 | 52 | 48 | 27 | 201 | 541 |
| Max | 148 | 150 | 172 | 81 | 57 | 59 | 61 | 57 | 34 | 224 | 700 |

**Table 5.** Statistical summary of 24-month age cows.

| | Chego (cm) | Sibza (cm) | Chejang (cm) | Hungsim (cm) | Hungpok (cm) | Yogak (cm) | Gojang (cm) | Gonpok (cm) | Jogol (cm) | Hungwe (cm) | Weight (kg) |
|---|---|---|---|---|---|---|---|---|---|---|---|
| Mean | 136.937 | 137.843 | 158.914 | 76.848 | 50.985 | 50.288 | 53.036 | 50.231 | 27.653 | 217.597 | 652.760 |
| STD | 4.256 | 4.278 | 7.166 | 3.107 | 3.936 | 3.193 | 3.618 | 3.608 | 3.088 | 9.225 | 66.245 |
| Min | 124 | 125 | 136 | 66 | 39 | 41 | 42 | 39 | 20 | 189 | 458 |
| Q1 | 134 | 135 | 154 | 75 | 48 | 48 | 50 | 48 | 25 | 211 | 606 |
| Q2 | 137 | 138 | 159 | 77 | 51 | 50 | 53 | 50 | 27 | 217 | 648 |
| Q3 | 140 | 141 | 164 | 79 | 54 | 52 | 56 | 53 | 30 | 224 | 696 |
| Max | 150 | 151 | 181 | 88 | 63 | 60 | 64 | 61 | 37 | 246 | 865 |

### 2.3. Machine Learning-Based Predictive Models

In this work, one of the major goals is to investigate the performance of supervised ML algorithms in estimating Hanwoo cow live weight. Currently, there are a huge number of supervised ML algorithms. Therefore, an exhaustive investigation considering all of the algorithms is not feasible in the scope of this research. As a result, three representative algorithms are taken into consideration, including Light Gradient Boosting Machine (LightGBM) [27], TabNet [28], and FT-Transformer [29]. In tabular data, tree-based ensemble learning is still believed to outperform other types of learning algorithms [24,25]. Nowadays, among various types of tree-based machine learning algorithms that have been proposed, LightGBM is considered one of the most efficient algorithms [27]. Whiles, DNN models are extensively employed for unstructured data. Recently researchers have been attempting to use DNN models, the most prominent are TabNet and FT-Transformer, for solving tabular data tasks.

Besides three modern ML models, kNN and MLP are two traditional ML algorithms taken into consideration to make a comparison. Among the five models using in this work, TabNet and FT-Transformer are DNN models, whiles kNN, MLP, and LightGBM are shallow ML models. In order to evaluate the performance of weight estimation, two metrics are exploited including Root Mean Squared Error (RMSE) and Mean Absolute Percentage Error (MAPE).

#### 2.3.1. Machine Learning Models

LightGBM is a Gradient Boosting Decision Tree (GBDT) algorithm invented by Ke et al. [27]. LightGBM incorporated two novel techniques: Gradient-based One-Side Sampling (GOSS) and Exclusive Feature Bundling (EFB). GOSS help to exclude a significant proportion of data samples with small gradients and keep the remaining data samples for estimating information gain. EFB helps to bundle mutually exclusive features to reduce the number of features. In this work, the LightGBM Python Package (version 3.3.2) was used to build the model with 100 base learners (decision tree) and the maximum tree depth of base learners is 32.

MLP is a conventional NN whose parameters are updated by the back-propagation training process [30]. MLPs are universal function approximators as shown by Cybenko's theorem [31]. Model kNN is a non-parametric ML algorithm since it does not make any assumptions on the data [32]. kNN algorithm uses feature similarity to predict the target values of new samples. This means that the target value of a new sample is computed by its distances to the data samples in the training dataset. Model MLP composes of two hidden layers with 30 and 20 neurons, respectively; and model kNN with $k = 11$ were built with the the Scikit-Learn Python Package (version 1.0.2).

#### 2.3.2. Deep Neural Network Models

TabNet is a DNN model employing the attention mechanism for tabular data invented by Sercan O Ark and Tomas Pfister in 2021 [28]. A TabNet model consists of an encoder component and a decoder component as shown in Figure 2. The encoder component composes of a feature transformer, an attentive transformer, a feature masking, a split block, and a Rectified Leaky Unit (ReLU) layer. The decoder component composes of a feature transformer and a Fully-connected Layer (FC) in each step. In this work, the model TabNet was built with Pytorch-Tabnet Python Package (version 4.0). To train the model with back-propagation training, Adam optimization algorithm [33] was adopted with parameters *learning rate* $= 0.02$, *betas* $= (0.9, 0.999)$, and *epsilon* $= 10^{-15}$.

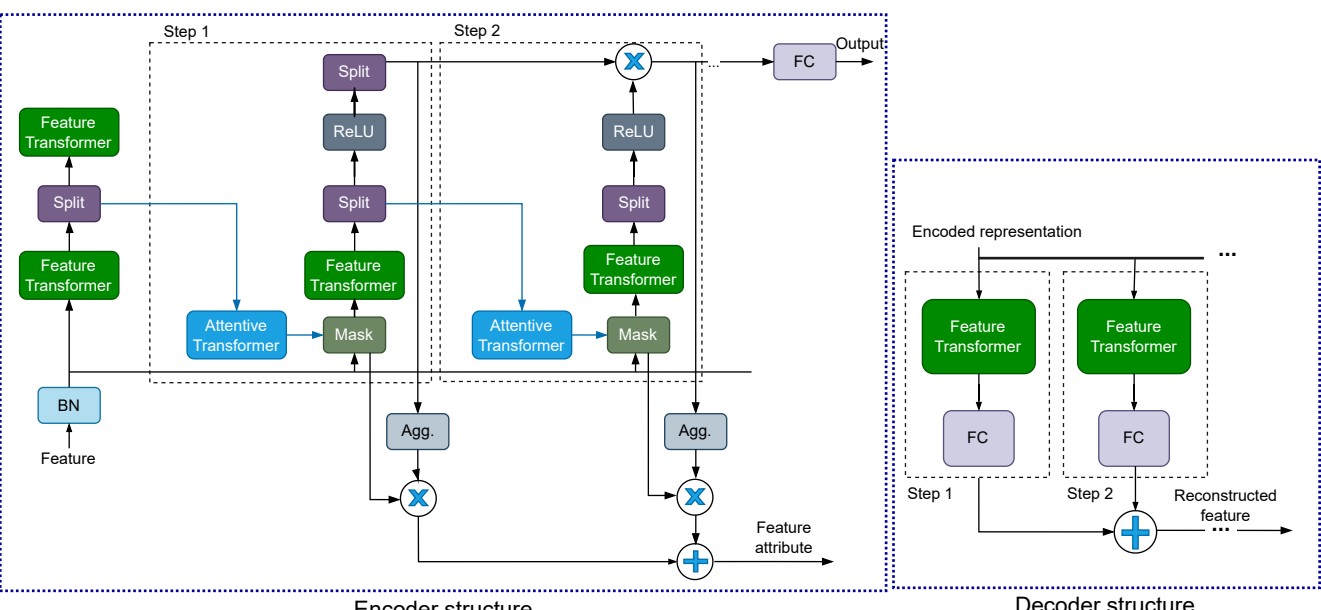

**Figure 2.** Architecture of TabNet [28].

FT-Transformer was invented by Gorishniy Y. et al. [29] in 2021 is a simple and efficient adaptation of transformer architecture-based for the tabular data. The synonym FT-Transformer stands for Feature Tokenizer Transformer. The architecture of FT-Transformer consists of Feature Tokenizer module and Transformer module as shown in Figure 3. Feature Tokenizer module transforms the input features $x$ into embedding $T$. After tokenizing, the stacked embedding $T_0$ is obtained by stacking the embedding $T$ token $[CLS]$. Transformer layers $F_1, \ldots, F_L$ are applied to obtain $T_i$, where $T_i = F_i(T_{i-1})$. In this work, model FT-Transformer was built with RTDL Python Package (version 0.0.13) [29]. To train the model with back-propagation training, AdamW optimization algorithm [33] was adopted with parameters *learning rate* = 0.001, *betas* = (0.9, 0.999), *epsilon* = $10^{-8}$, and *weight decay* = 0.01.

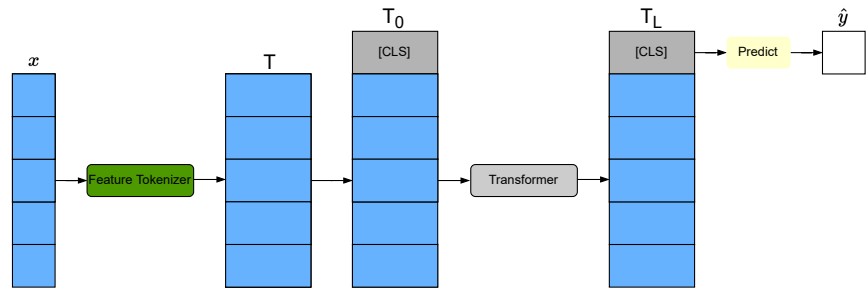

**Figure 3.** Architecture of FT-Transformer [29].

## 3. Results

### 3.1. Correlation Analysis

Pearson correlation [34] is employed to measure the linear correlation between two body size measurements. The correlation results are shown in Table 6. All the corresponding p-values are also computed and showed very small values which indicate that all measurements have significant correlations with the live weight of the cattle body. Of all body measurements, Hungwe has the highest correlation with body weight under all ages of cows. The analysis also shows that all body size measurements highly correlate with each other, as in Figure 4.

**Table 6.** Pearson correlation between body measurements and body weight.

| Feature | 6-Month Cow | 12-Month Cow | 18-Month Cow | 24-Month Cow | STD |
|---|---|---|---|---|---|
| Chego | 0.695 | 0.627 | 0.669 | 0.559 | 0.059 |
| Sibza | 0.720 | 0.680 | 0.666 | 0.551 | 0.073 |
| Chejang | 0.736 | 0.799 | 0.753 | 0.703 | 0.040 |
| Hungsim | 0.558 | 0.699 | 0.707 | 0.670 | 0.069 |
| Hungpok | 0.464 | 0.708 | 0.676 | 0.569 | 0.111 |
| Yogak | 0.474 | 0.628 | 0.700 | 0.616 | 0.095 |
| Gojang | 0.520 | 0.619 | 0.577 | 0.490 | 0.058 |
| Gonpok | 0.466 | 0.710 | 0.682 | 0.662 | 0.111 |
| Jagol | 0.295 | 0.544 | 0.650 | 0.612 | 0.160 |
| Hungwe | 0.896 | 0.913 | 0.923 | 0.904 | 0.012 |

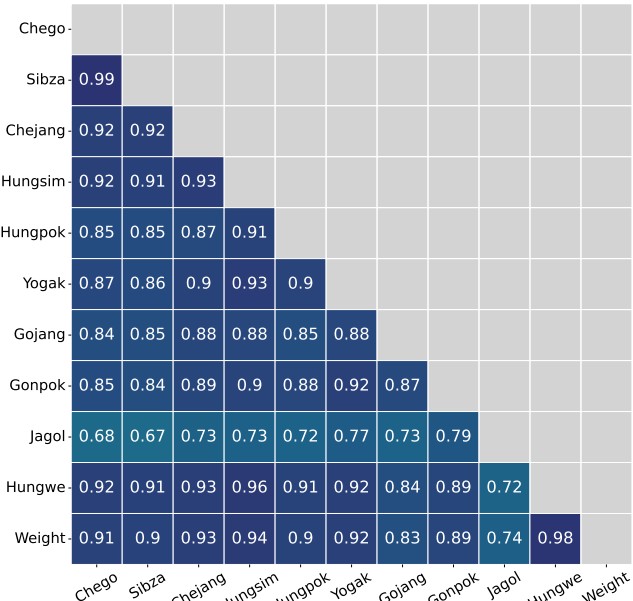

**Figure 4.** Correlation values between body measurements.

It can be observed that Hungpok, Yogak, Gonpok, and Jagol change their correlation with live weight a lot among four categories of months. For example, in 6-month data, the correlation value between Jagol feature and weight is only 0.295, while in 18-month age data the correlation value is 0.65. Based on that observation, body size measurements are organized into three groups: group A is of stable correlation variables (Hungwe, Chego, Sibza, Chejang, Hungsim, and Gojang); group B includes unstable correlation variables (Hungpok, Yogak, Gonpok, and Jagol), and group C consists of all variables.

*3.2. Live Weight Prediction Results*

Each considered ML model will be developed with three different groups of variables: A, B, and C. Moreover, the age values of individuals are also considered an independent variable and used as input features to train predictive models. The training and validation datasets are used during the training process. The training dataset is used to update the model parameters while the validation dataset is used to stop the training process early to prevent overfitting. The test dataset is reserved for evaluating trained models. All final evaluation results computed on the test dataset are shown in and Figure 5.

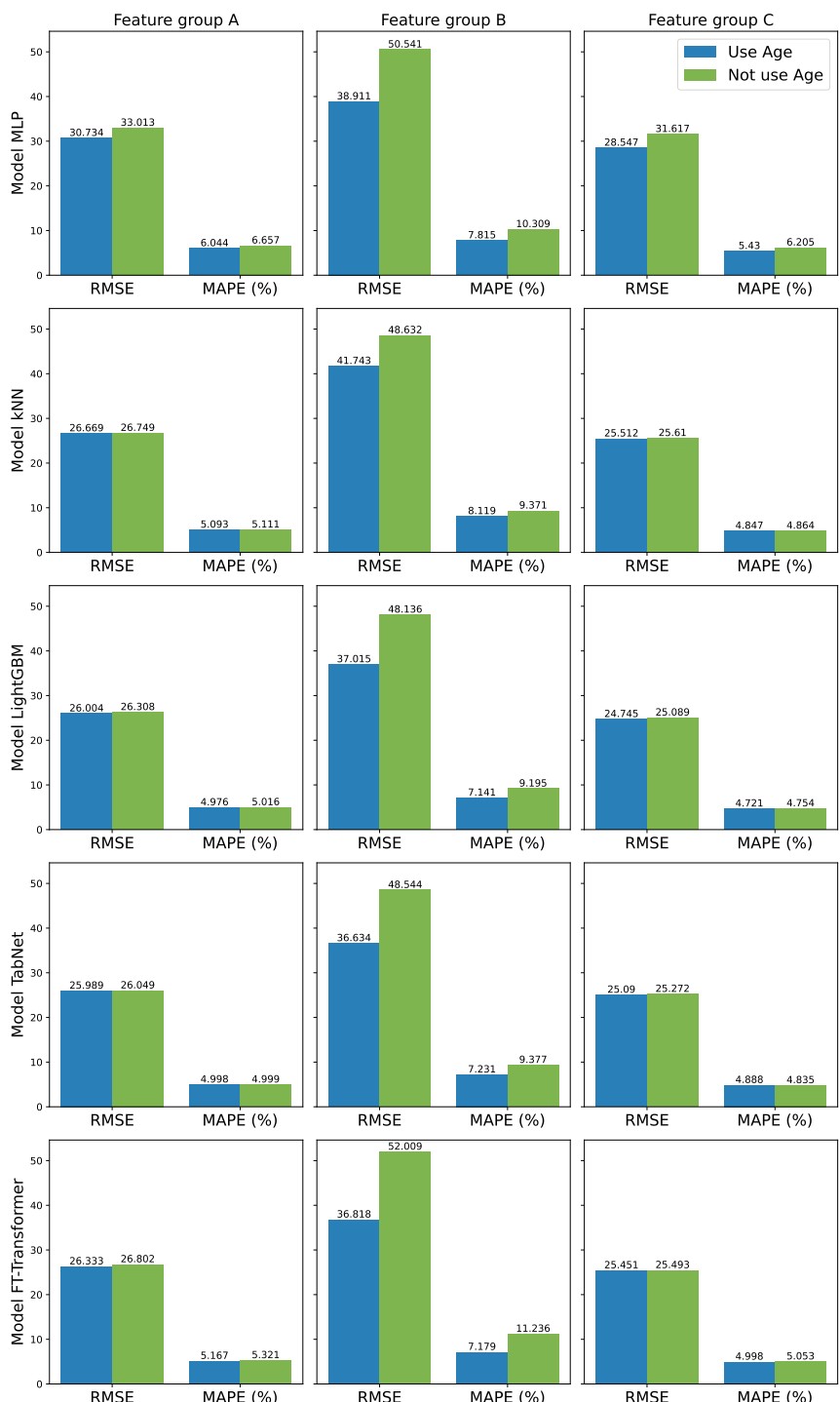

**Figure 5.** Estimation errors (RMSE and MAPE) of five models using different features.

Among three groups of features A, B, and C, it can be observed that group B which consists of unstable features (Hugok, Yogak, Gonpok, Jagol) has the poorest performance. Group A which consists of stable features (Chego, Sibza, Chejang, Hungsim, Hungpok, Yogak, Gojang, Gonpok, Jagol, Hungwe) has better performance. Group C which includes all features gives the best performance of all groups.

Compared between the two cases of using and not using age as an input feature, it can be observed that when a model uses age as an input feature, it often has lower RMSE and MAPE values. In the cases of feature groups A and C, the differences between using age and not using age are very small. However, in the case of feature group B, the difference is

quite obvious. For example, in the FT-Transformer case, using age can have 36.818% and 7.179% for RMSE and MAPE, respectively. But if not using age, the performance decreases dramatically to 52.009% and 11.236% for RMSE and MAPE, respectively.

Among the five compared models, LightGBM has the best performance. After considering all combinations of cases, it can be noted that the best performance is in the case of using 10 features, which means 10 body size measurements without age, with model LightGBM (RMSE = 24.754, MAPE = 4.721%). The worst case is the case with the FT-Transformer model, using only four features Hungpok, Yogak, Gonpok, and Jagol; in this case, RMSE = 52.009, MAPE = 11.236%.

In order to attenuate the effect of the randomness in analyzing the results of predictions, experiences with each model are conducted 6 times with the random initialization of parameters and split of training set—validation set. Dispersion analysis of RMSE and MAPE errors are shown in Table 7 and Figure 6. In all the cases, it can be observed that the kNN model has the smallest dispersion and the MLP model has the biggest dispersion. The reason explaining for that small dispersion value is that kNN is a non-parametric model. kNN is not affected by the random initialization of parameters. In general, all models but MLP have small dispersion which indicates that the prediction results of these models are reliable.

**Table 7.** Dispersion analysis of RMSE and MAPE errors in case of using all features.

| Model | Features | RMSE | MAPE(%) |
|---|---|---|---|
| MLP | | 28.547 ± 0.263 | 5.430 ± 0.091 |
| kNN | Use all features | 25.512 ± 0.096 | 4.847 ± 0.017 |
| LightGBM | and use age as | 24.745 ± 0.135 | 4.721 ± 0.023 |
| TabNet | another feature | 25.090 ± 0.388 | 4.888 ± 0.185 |
| FT-Transformer | | 25.451 ± 0.956 | 4.998 ± 0.212 |
| MLP | | 31.617 ± 3.911 | 6.205 ± 0.777 |
| kNN | Use all features | 25.610 ± 0.049 | 4.864 ± 0.008 |
| LightGBM | (not use age) | 25.089 ± 0.072 | 4.754 ± 0.019 |
| TabNet | | 25.272 ± 0.243 | 4.835 ± 0.069 |
| FT-Transformer | | 25.493 ± 0.192 | 5.053 ± 0.053 |

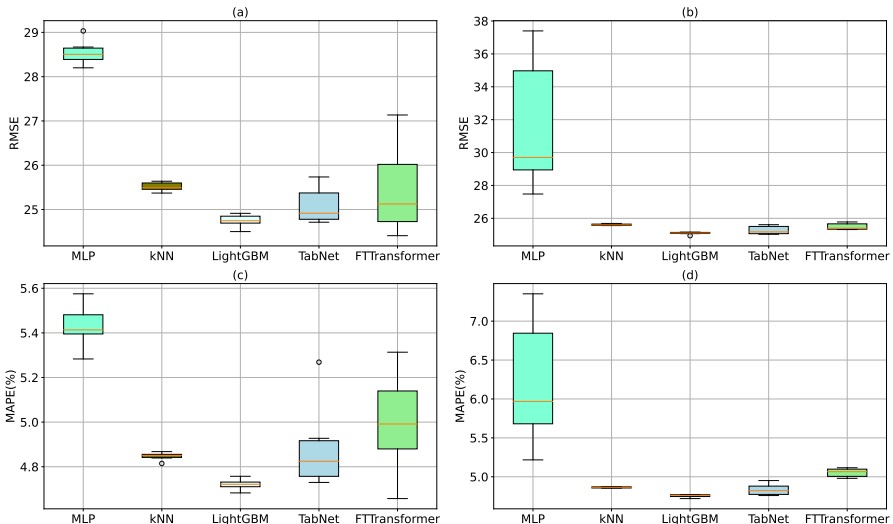

**Figure 6.** Dispersion analysis of RMSE and MAPE errors in case of using all features; (**a**,**c**): Using age as a feature; (**b**,**d**): Not using age.

### 3.3. Feature Importance

The feature importance of body measurements according to the LightGBM model in the case of using all features is shown in Figure 7. It can be observed that Hungwe is

the most important feature while Jagol is the least important feature. The second most important feature is Chejang. All other features have lower and similar importance but still have a large contribution to the prediction result of the LightGBM model. This result indicates that all features should be included in the predictive models.

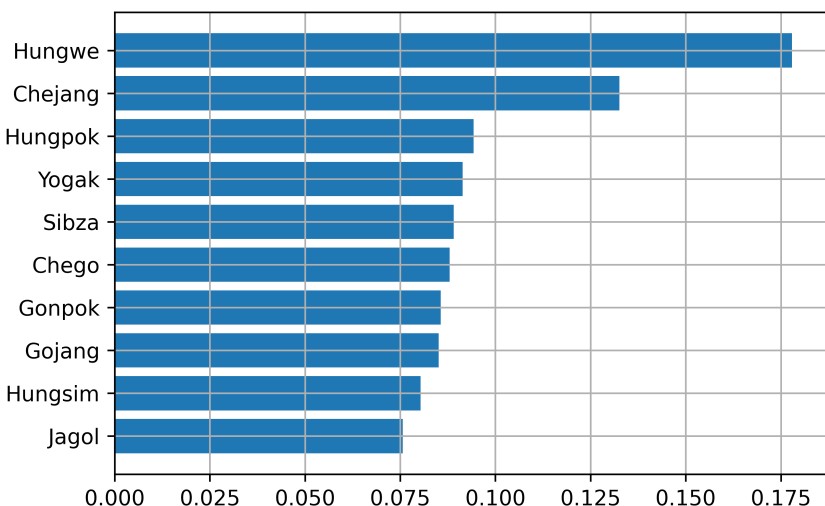

**Figure 7.** Feature importance according to LightGBM model.

## 4. Conclusions

In this work, ten body measurements of Hanwoo cow were used as the input features for the estimation of the body live weight. Pearson correlation analysis showed that all of the body measurement has the high correlation with the body weight. Among all features, the girth of chest girth (Hungwe) has the highest correlation with the body weight, while the width of hip bone (Jagol) has lowest correlation with the body weight. Experiment results showed that using different sets of features affects the performance of weight estimation. Using all features together provided the best performance in all cases of the estimation models. Age value was used as another feature to estimate body weight, and that often give a slightly better results in most case. Five different ML models have been investigated and evaluated. The tree-based model LightGBM regression demonstrated the highest performance. The results of this work will be used to develop an indirect live weight estimation for Hanwoo, in which machine vision technology is utilized to automatically measure ten body features of cows.

**Author Contributions:** Conceptualization, C.D. and D.H.; methodology, D.H. and S.H.; software, D.H.; validation, M.P., S.L. (Seungsoo Lee), S.L. (Soohyun Lee) and M.A.; formal analysis, M.A. and J.L.; investigation, C.D. and M.P.; resources, S.L. (Seungsoo Lee), S.L. (Soohyun Lee) and M.A.; data curation, D.H., S.L. (Seungsoo Lee), S.L. (Soohyun Lee) and M.A.; writing—original draft preparation, D.H. and S.H.; writing—review and editing, D.H. and S.H.; visualization, S.H.; supervision, C.D., T.C., S.L. (Seungsoo Lee) and S.L. (Soohyun Lee); project administration, C.D. and M.A.; funding acquisition, C.D., T.C., M.P., S.L. (Seungsoo Lee), S.L. (Soohyun Lee), M.A. and J.L. All authors have read and agreed to the published version of the manuscript.

**Funding:** This work was supported by Korea Institute of Planning and Evaluation for Technology in Food, Agriculture and Forestry (IPET) and Korea Smart Farm R&D Foundation (KosFarm) through Smart Farm Innovation Technology Development Program, funded by Ministry of Agriculture, Food and Rural Affairs (MAFRA) and Ministry of Science and ICT (MSIT), Rural Development Administration (421050-03).

**Institutional Review Board Statement:** The animal study protocol was approved by the IACUC at National Institute of Animal Science (approval number: NIAS20222380).

**Informed Consent Statement:** Not applicable.

**Data Availability Statement:** Not applicable.

**Conflicts of Interest:** The authors declare no conflict of interest.

## Abbreviations

The following abbreviations are used in this manuscript:

| | |
|---|---|
| STD | Standard Deviation |
| Q1 | Quartile 1 |
| Q2 | Quartile 2 |
| Q3 | Quartile 3 |
| kNN | k-Nearest Neighbor |
| FT-Transformer | Feature Tokenizer Transformer |
| NN | Neural Network |
| ANN | Artificial Neural Network |
| CNN | Convolutional Neural Network |
| DNN | Deep Neural Network |
| ML | Machine Learning |
| DL | Deep Learning |
| LightGBM | Light Gradient Boosting Machine |
| RMSE | Root Mean Squared Error |
| MAPE | Mean Absolute Percent Error |

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
