# Peer review of "Machine Learning-Based Live Weight Estimation for Hanwoo Cow"

_sustainability, doi:10.3390/su141912661_

Round 1

Reviewer 1 Report

The papers deals with a very important issue today, which is the estimation of indirect productivity of cattle. However, I suggest improvements in several aspects so that it can be effectively published in the journal.

Materials and methods:

- Tables 2 to 5 have labels described in a very simple way, making it difficult for the reader to interpret. The measurement units used were also not mentioned.

- The authors do not justify at any time the choice of the three algorithms used and compared.

- I don't see the need to describe the algorithms used, the references are enough in this case. Because these are machine learning algorithms, it is better to describe the execution biases and heuristics than to simply describe each algorithm step by step.

- The evaluation metrics are well known in the literature and I also don't see the need to describe them in the paper, just the references.

Results: 

  -It was not described which correlation measure was used to verify the relationships between the variables. Was a simple Pearson correlation used? Or a more sophisticated measure?

- The graphs in Figure 3 and 4 are very similar. What justifies the use of the graphs in Figure 4? Just to show the differences using or not using the age attribute?

- I believe that a single figure can be made showing these results. Information is also missing on the label of the figures.

Conclusions:

- Conclusion is very vague, leaving open many questions that could be clarified in the article. The data are from a very specific breed of animals, under unreported climatic and management conditions, meaning that the result cannot be used immediately for cattle under any circumstances.

- Future works, important for the continuity of the research, are not cited. But they could be suggested from a more in-depth analysis of what was done in this specific paper.

Finally, the dataset used in this paper is apparently quite rich, with information collected in the field, and could be made available by the authors for further research.

Reviewer 2 Report

The manuscript proposes the use of ML models to estimate the live weight of Hanwoo cattle based on

their measurements. 

Some remarks:

1- The title presents an error: "ML-based FOR Hanwoo Cattle Live Weight Estimation"

2- Abstract: the authors should mention the ML algorithms used in the manuscript;

3- Line 28: The 2nd paragraph is too large. Please, split it;

4- Line 70: "ANN is a ML algorithm that utilizes data computational structure miming human brain 

neural cells" -> Indeed, the ANNs were inspired by the functioning of the nervous system of the superior

organisms. They do not mimic the brain.

5- The authors must mention the ML models addressed in Introduction;

6- The acronym LightGBM must be defined after the appearance of the full name;

7- The description of the models is too shallow. Figures, pseudocodes, and details must be provided;

8- Eq. 2: asterisk means convolution. Please, correct;

9- In table 6, the first value presents an inconsistency;

10- The x axis of KNN and LightGBM must be defined in all subfigures;

11- The authors did not mention how many times they ran the stochastic algorithms. Due to the random

initialization, the tunning usually found distinct values to the parameters;

12- In this sense, a dispersion analysis is required. The authors may use the boxplot;

13- It must be clear why the authors selected those 4 models. Also, I suggest the application of some

traditional neural model, such as the MLP;

14- Conclusion must be Section 4;

15- The conclusion is too small and does not present the structure expected.

Round 2

Reviewer 1 Report

New version well written and adjusted as requested. No further adjustments are required.

Reviewer 2 Report

The authors much improved the manuscript. I accept it for publication on Sustainability